# Sample Size and Estimation of Standard Radiation Doses for Pediatric Brain CT

**Yusuke Inoue** [1,*] **, Hiroyasu Itoh** [2] **, Nao Shiibashi** [1] **, Ryosuke Sasa** [2] **and Kohei Mitsui** [1]

1   Department of Diagnostic Radiology, Kitasato University School of Medicine, Sagamihara 252-0374, Kanagawa, Japan
2   Department of Radiology, Kitasato University Hospital, Sagamihara 252-0374, Kanagawa, Japan
*   Correspondence: inoueys@kitasato-u.ac.jp

**Abstract:** Estimation of the standard radiation dose at each imaging facility is required for radiation dose management, including establishment and utilization of the diagnostic reference levels. We investigated methods to estimate the standard dose for pediatric brain computed tomography (CT) using a small number of data. From 980 pediatric brain CT examinations, 25, 50, and 100 examinations were randomly extracted to create small, medium, and large datasets, respectively. The standard dose was estimated by applying grouping and curve-fitting methods for 20 datasets of each sample size. For the grouping method, data were divided into groups according to age or body weight, and the standard dose was defined as a median value in each group. For the curve-fitting methods, logarithmic, power, and bilinear functions were fitted to plots of radiation dose against age or weight, and the standard dose was calculated at the designated age or weight using the derived equation. When the sample size was smaller, the random variations of the estimated standard dose were larger. Better estimation of the standard dose was achieved with the curve-fitting methods than with the grouping method. Power fitting appeared to be more effective than logarithmic and bilinear fittings for suppressing random variation. Determination of the standard dose for pediatric brain CT by the curve-fitting method is recommended to improve radiation dose optimization at facilities performing the imaging procedure infrequently.

**Keywords:** computed tomography; radiation dose; pediatrics; brain; diagnostic reference level; sample size

## 1. Introduction

For medical imaging utilizing ionizing radiation, the potential detrimental effects of radiation exposure—especially the increased risk of cancer—are an important consideration. Following the principle of optimization, the radiation dose to the patients should be minimized while maintaining the clinical benefits. Computed tomography (CT) is a major source of radiation exposure. Children are sensitive to radiation, and their long expected lifetimes allow the development of cancer after a long latency period [1]. Accordingly, optimizing the radiation dose for pediatric CT is a priority. In children, CT is most frequently used for brain imaging [2,3], and an increased incidence of brain tumors has been reported in children who underwent brain CT [4–6]. Therefore, optimization of the radiation dose in pediatric brain CT is of particular importance.

The use of diagnostic reference levels (DRLs) is recommended to promote optimization of radiological imaging [7,8]. In establishing and utilizing DRLs, first, each imaging facility investigates radiation dose indices recorded in clinical practice to determine the standard dose at the facility. Usually, the standard dose is defined as the median value for 30 or more data obtained from imaging standard-sized patients. The national authority surveys standard doses at many facilities in the country and determines DRLs based on the distribution of those doses. Typically, the 75th percentile value of the distribution is

defined as the DRL. Each facility then compares its standard dose to the DRL. A standard radiation dose higher than the DRL indicates that the facility uses a relatively high dose in comparison with those used at other facilities in the country, such that a dose reduction is likely to be achievable.

The amount of radiation exposure required to obtain diagnostic-quality CT images depends on the strength of X-ray attenuation by the imaging sections. Imaging of larger sections, causing stronger attenuation, requires more radiation exposure. Because children vary widely in size, the optimal dose is variable, and standard doses should be determined according to the patient's size. It has been recommended to divide children into groups based on age and body weight for brain and body CTs, respectively, and to define the median value in a given age or weight group as the DRL [8,9].

However, the need for grouping based on age or weight causes difficulties in radiation dose management for pediatric CT. In general, CT is not frequently performed in children because of its high radiation dose. The number of pediatric CT examinations is small, and grouping makes the number in each group smaller, disturbing the determination of a median value with acceptable statistical validity. The section size may be variable within an age group (e.g., <1 year) or weight group (e.g., 5–15 kg), and estimation of the standard dose becomes susceptible to a deviation in age or weight within a group when the number of data is small. Although the use of data from at least 10 CT examinations per group has been recommended to determine the standard dose at a given facility [9], it is difficult for many facilities to collect a sufficient number of data [10]. When following the recommendations, DRLs are established based on the standard doses at facilities where pediatric CT examinations are frequently performed; therefore, they may not comprehensively represent the typical radiation dose in the country. In a previous study based on the American College of Radiology's Dose Index Registry, half of the facilities included in the analysis performed fewer than 10 CT examinations per month in patients aged ≤ 18 years (median age of 14 years) [2]. The number of examinations should have been much smaller for a given body region in young children. Because the radiation dose for pediatric CT has been shown to be higher in non-pediatric facilities than in pediatric facilities [11–13], CT in non-pediatric facilities with less experience of pediatric CT should be an important target for optimization. It is desired to survey the standard radiation dose extensively, including at facilities where pediatric CT is infrequently performed, to establish DRLs representing the radiation dose in the country comprehensively and to encourage each facility to optimize their CT practice with reference to the DRLs.

To overcome the sample size problem and facilitate standard dose determinations at facilities performing a small number of pediatric CT examinations, curve-fitting of plots of radiation dose indices against weight, using weight as a continuous variable, has been proposed [14,15] and recommended for body CT [8,9]. Determination of standard doses using curve-fitting has also been reported for pediatric brain CT using a large number of CT data [16], and the study indicated the superiority of weight over age as a variable and that of a bilinear function over logarithmic and power functions as a fitting function. In this study, we examined the different methods used to determine standard doses in pediatric brain CT in the context of a small sample size. Standard doses were estimated from many small subsamples of CT data extracted randomly from a larger dataset, and variations therein were evaluated. The principal aim of this study was to aid in standard dose determination and radiation dose optimization for pediatric brain CT at facilities where this imaging procedure is infrequently performed.

## 2. Materials and Methods

### 2.1. Subjects

Brain CT examinations performed in children aged < 15 years at a single institution were retrospectively analyzed. The data were used in a previous study [16] and were reanalyzed for different purposes. The study protocol was approved by Kitasato University's Medical Ethics Organization (Sagamihara, Japan) (B20-114), and the need for

informed consent was waived. In patients who underwent two or more CT examinations, those performed at an interval > 1 year were analyzed. The exclusion criteria were as follows: lack of data on weight (*n* = 52), weight > 80 kg (*n* = 5), helical-mode imaging (*n* = 8), imaging using an adult protocol (*n* = 9), and no use of a head holder (*n* = 5). Finally, 980 examinations (544 males and 436 females) were considered eligible.

Patient age was calculated as the difference in years, months, and days between the date of birth and date of examination, and was expressed in years. When CT was performed on the day of birth, age was regarded as 0.003 years (equal to 1 day) for logarithmic transformation.

## 2.2. Imaging Procedures

The CT imaging procedures were described previously [16]. Briefly, two 64-detector-row CT scanners with the same specifications (Optima CT 660 Discovery Edition; GE Healthcare, Milwaukee, WI, USA) were used. After acquiring posteroanterior and lateral localizer images, axial CT images parallel to the orbitomeatal line were obtained in non-helical mode, covering the posterior fossa and the top of the brain. Tube current was determined by automatic exposure control (AEC) software—i.e., Auto mA and Smart mA (GE Healthcare)—to modulate radiation exposure according to the X-ray attenuation for each patient and at each imaging location [17]. The noise index was set to 4. Organ dose modulation was applied to the orbital region to reduce the radiation dose to the eye lens. [18,19]. Other imaging parameters were as follows: tube voltage, 120 kV; rotation time, 1 s; beam width, 10 mm; slice thickness, 5 mm; and slice increment, 5 mm. The volume CT dose index (CTDIvol) provided by the CT scanner mentioned above was recorded and analyzed as an index of the radiation dose.

## 2.3. Age and Weight Division for Bilinear Fitting

In bilinear curve fitting, patient age or weight is divided into two ranges at a predefined cutoff, and a linear regression equation is determined for each range. We assessed the effect of the cutoff value on bilinear fitting. When age-based analysis was performed with a cutoff of 1 year, CTDIvol was plotted against age, and linear regression analysis was performed separately for the age ranges 0–<1 year (young range) and 1–<15 years (old range). The age at each examination was substituted into the two resulting regression equations, and CTDIvol estimated from age was derived as the smaller of the two values. Similarly, CTDIvol was estimated using cutoffs of 1.5 and 2 years. Estimation was also performed by monolinear fitting of data from all 980 examinations. The error of estimation (%) was defined as follows: (estimated value − actual value)/(actual value) × 100. Mean errors were calculated at 0–<0.25, 0.25–<1, 1–<5, 5–<10, and 10–<15 years. In the weight-based analysis, the weight range was divided into the light and heavy ranges at cutoffs of 10, 15, and 20 kg. Mean errors of estimation were calculated at 0–<5, 5–<15, 15–<30, 30–<50, and 50–<80 kg.

## 2.4. Sample Size and Standard Dose Estimation

From the 980 CT examinations, 25, 50, or 100 examinations were extracted randomly to constitute small, medium, and large datasets, respectively (Figure 1). Random sampling was repeated 20 times to create 20 datasets of each size, using the RAND and RANK functions in Microsoft Excel (Microsoft Corp., Redmond, WA, USA).

Standard doses were estimated by applying the curve-fitting and grouping methods to each dataset. For the curve-fitting method, CTDIvol was plotted against age or weight, and logarithmic, power, and bilinear functions were fitted to the plots using Microsoft Excel, to determine equations for CTDIvol estimation. The cutoff in bilinear fitting was set to 1.5 years or 15 kg. Standard doses at 0.25, 1, 5, 10, and 15 years and at 3, 10, 20, 40, and 60 kg were calculated by substituting the age or weight value into the equation. For the grouping method, the standard dose was determined as the median of CTDIvol for the 0–<0.25, 0.25–<1, 1–<5, 5–<10, and 10–<15 years age groups and for the 0–<5, 5–<15, 15–<30, 30–<50, and 50–<80 kg weight groups. The standard dose for an age or weight

group was determined even when only one datum was available for the group. Among the 20 datasets of a given sample size, the mean (mGy) and coefficient of variation (CV, %) of the standard dose were computed.

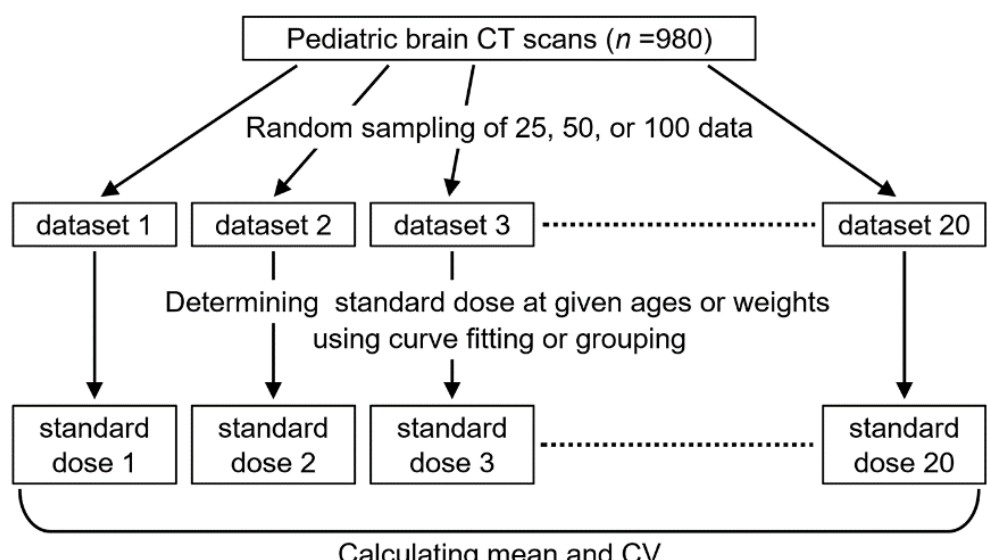

**Figure 1.** Flowchart of the estimation of standard doses from datasets of different sizes.

## 3. Results

### 3.1. Age and Weight Division for Bilinear Fitting

Irrespective of the age range, CTDIvol was correlated positively with age (Figure 2, Table 1). The slope of the regression line and the correlation coefficient were lower in the old range than in the young range. For the young range, the slope decreased with increasing cutoff age. The effect of the cutoff age was negligible for the old range.

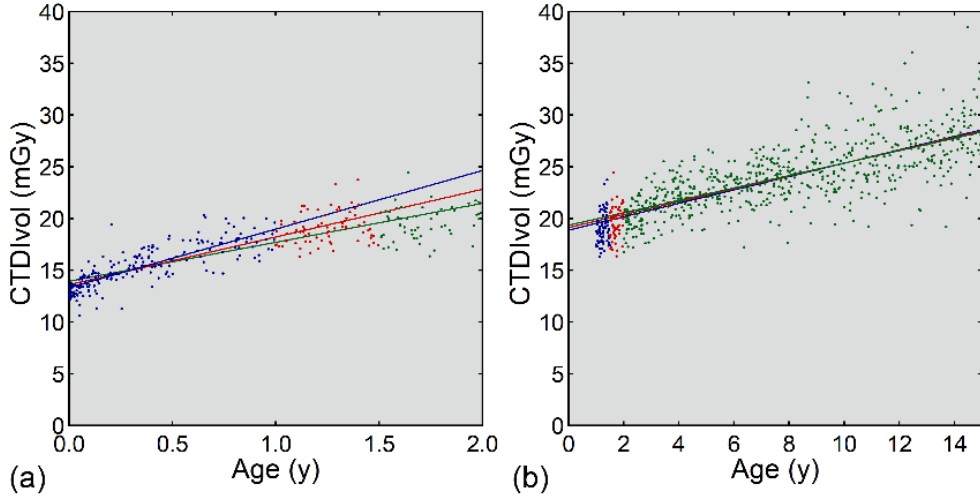

**Figure 2.** Linear regression analysis of CTDIvol and age for <2 years (**a**) and ≥1 year (**b**). In panel (**a**), the blue, red, and green plots represent data of 0–<1, 1–<1.5, and 1.5–<2 years, respectively. In panel (**b**), the blue, red, and green plots represent data of 1–<1.5, 1.5–<2, and 2–<15 years, respectively. The blue, red, and green lines are regression lines determined using young/old cutoffs of 1, 1.5, and 2 years, respectively. For example, with a cutoff of 1.5 years, linear regression was performed using the blue and red plots in the young range (**a**) and the red and green plots in the old range (**b**).

**Table 1.** Age range and results of linear regression between CTDIvol and age.

| Age (y) | Equation | r | n |
|---|---|---|---|
| 0–<1 | y = 5.671x + 13.35 | 0.830 | 211 |
| 0–<1.5 | y = 4.620x + 13.62 | 0.862 | 283 |
| 0–<2 | y = 3.777x + 13.94 | 0.849 | 339 |
| 1–<15 | y = 0.648x + 18.89 | 0.788 | 769 |
| 1.5–<15 | y = 0.625x + 19.13 | 0.753 | 697 |
| 2–<15 | y = 0.601x + 19.37 | 0.716 | 641 |
| 0–<15 | y = 0.848x + 16.98 | 0.848 | 980 |

Similarly, CTDIvol was correlated positively with weight (Figure 3, Table 2), and the slope of the regression line and the correlation coefficient were lower in the heavy range than in the light range. For both ranges, the slope decreased and the y-intercept increased with increasing cutoff weight.

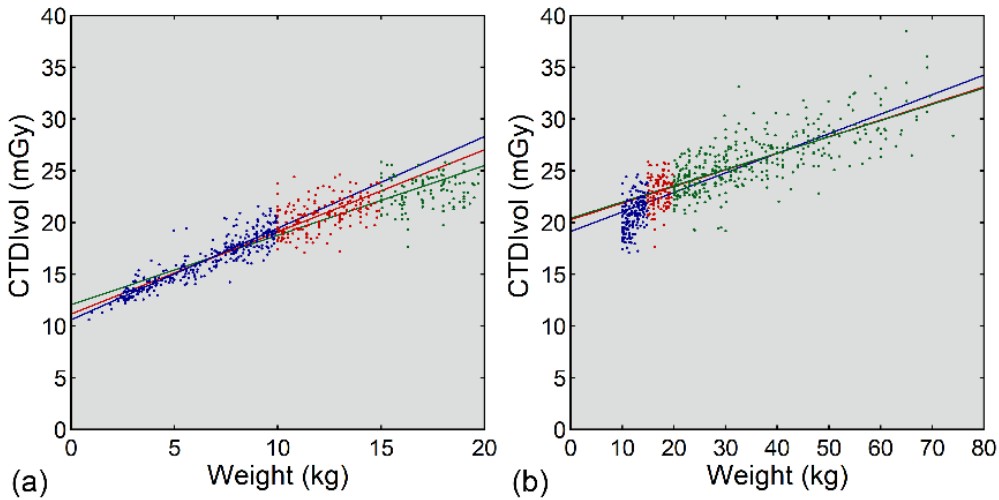

**Figure 3.** Linear regression analysis of CTDIvol and weight for < 20 kg (**a**) and ≥ 10 kg (**b**). In panel (**a**), the blue, red, and green plots represent data of 0–<10, 10–<15, and 15–<20 kg, respectively. In panel (**b**), the blue, red, and green plots represent data of 10–<15, 15–<20, and 20–<80 kg, respectively. The blue, red, and green lines are regression lines determined using light/heavy cutoffs of 10, 15, and 20 kg, respectively.

**Table 2.** Weight ranges and results of linear regression between CTDIvol and weight.

| Weight (kg) | Equation | r | n |
|---|---|---|---|
| 0–<10 | y = 0.8854x + 10.62 | 0.912 | 299 |
| 0–<15 | y = 0.7941x + 11.17 | 0.920 | 490 |
| 0–<20 | y = 0.6729x + 12.07 | 0.915 | 608 |
| 10–<80 | y = 0.1894x + 19.14 | 0.830 | 681 |
| 15–<80 | y = 0.1609x + 20.28 | 0.766 | 490 |
| 20–<80 | y = 0.1580x + 20.41 | 0.724 | 372 |
| 0–<80 | y = 0.2696x + 16.32 | 0.862 | 980 |

In the estimation of CTDIvol based on age, the mean error at 0–<0.25 years was much higher than 0 when monolinear fitting was used, implying severe overestimation of the radiation dose (Table 3). The mean error was closer to 0 when using bilinear fitting than when using monolinear fitting, irrespective of age, indicating better estimation. Of the three cutoff ages, the mean error was closest to 0 for a cutoff of 1 year. The use of 1.5 years as the cutoff caused slight overestimation at 0–<0.25 years and slight underestimation at 0.25–<1 years. Such overestimation and underestimation were slightly increased when the cutoff was 2 years.

**Table 3.** Age cutoffs for bilinear fitting and error of CTDIvol estimation.

| Age Group (y) | Error (%) | | | | *n* |
|---|---|---|---|---|---|
| | **1 y** | **1.5 y** | **2 y** | **Monolinear** | |
| 0–<0.25 | 1.3 ± 7.0 | 2.7 ± 7.3 | 4.6 ± 7.6 | 25.0 ± 9.9 | 101 |
| 0.25–<1 | −0.1 ± 8.0 | −2.1 ± 7.7 | −3.1 ± 7.7 | 5.1 ± 9.8 | 110 |
| 1–<5 | 1.0 ± 8.0 | 1.3 ± 7.9 | 0.9 ± 8.1 | −5.8 ± 7.2 | 313 |
| 5–<10 | −0.5 ± 8.7 | −0.2 ± 8.7 | 0.0 ± 8.7 | −2.3 ± 8.9 | 239 |
| 10–<15 | 1.9 ± 10.9 | 1.7 ± 10.8 | 1.5 ± 10.8 | 4.0 ± 11.1 | 217 |

Values are presented as the mean ± SD; 1 y, 1.5 y, and 2 y indicate bilinear fitting with cutoffs of 1, 1.5, and 2 years, respectively.

In the weight-based estimation, monolinear fitting caused severe overestimation at 0–<5 kg (Table 4). With bilinear fitting, the mean error was closer to 0, while overestimation was relatively apparent at 0–<5 kg using the 20 kg cutoff and at 50–<80 kg using the 10 kg cutoff. Additionally, Table 4 shows that the number of examinations for the 50–<80 kg weight group was relatively small.

**Table 4.** Weight cutoffs for bilinear fitting and error of CTDIvol estimation.

| Weight Group (kg) | Error (%) | | | | *n* |
|---|---|---|---|---|---|
| | **10 kg** | **15 kg** | **20 kg** | **Monolinear** | |
| 0–<5 | −0.1 ± 4.9 | 1.8 ± 5.1 | 5.4 ± 5.6 | 26.9 ± 8.9 | 100 |
| 5–<15 | 0.9 ± 6.4 | 0.1 ± 6.5 | −1.4 ± 6.4 | 0.3 ± 9.3 | 390 |
| 15–<30 | −1.7 ± 7.0 | 0.7 ± 7.1 | 0.7 ± 7.2 | −6.5 ± 7.0 | 280 |
| 30–<50 | −0.2 ± 7.4 | 0.1 ± 7.4 | 0.1 ± 7.4 | 0.7 ± 8.0 | 155 |
| 50–<80 | 2.9 ± 8.4 | 1.1 ± 8.3 | 1.0 ± 8.2 | 9.3 ± 9.1 | 55 |

The 10 kg, 15 kg, and 20 kg indicate bilinear fitting with cutoffs of 10, 15, and 20 kg, respectively.

### 3.2. Sample Size and Standard Dose Estimation

Among the 20 small datasets, no data were available in 2 of 100 age groups (Table 5) and 5 of 100 weight groups (Table 6), such that median doses could not be determined for these 7 groups. One medium dataset had no data at 50–<80 kg. There were fewer than five data in many age or weight groups of the small datasets. All or most datasets had fewer than five data at 0–<0.25 and 0.25–<1 years and at 0–<5 and 50–<80 kg. Groups with fewer than five data decreased for the medium datasets. They further decreased for the large datasets; however, fewer than five data were available at 50–<80 kg in five large datasets.

**Table 5.** Age groups with a small number of data.

| Age Group (y) | No Data | | | <5 Data | | |
|---|---|---|---|---|---|---|
| | **S** | **M** | **L** | **S** | **M** | **L** |
| 0–<0.25 | 1 | 0 | 0 | 20 | 7 | 1 |
| 0.25–<1 | 1 | 0 | 0 | 18 | 7 | 0 |
| 1–<5 | 0 | 0 | 0 | 1 | 0 | 0 |
| 5–<10 | 0 | 0 | 0 | 5 | 0 | 0 |
| 10–<15 | 0 | 0 | 0 | 8 | 0 | 0 |

Values are the number of groups. S, M, and L represent small, medium, and large datasets, respectively.

**Table 6.** Weight groups with a small number of data.

| Weight Group (kg) | No Data | | | <5 Data | | |
|---|---|---|---|---|---|---|
| | **S** | **M** | **L** | **S** | **M** | **L** |
| 0–<5 | 2 | 0 | 0 | 18 | 8 | 1 |
| 5–<15 | 0 | 0 | 0 | 0 | 0 | 0 |
| 15–<30 | 0 | 0 | 0 | 2 | 0 | 0 |
| 30–<50 | 0 | 0 | 0 | 14 | 2 | 0 |
| 50–<80 | 3 | 1 | 0 | 20 | 14 | 5 |

The relationships between CTDIvol and weight for the small, medium, and large datasets are exemplified in Figure S1. The sample size did not affect the median values of the coefficients of determination between CTDIvol and age (Table S1) or weight (Table S2), whereas the coefficients of determination were exceptionally low in one small dataset. The standard doses estimated based on age or weight and averaged across the 20 datasets of a given size are shown in Figure 4. The mean standard dose did not vary consistently according to the size of the dataset for any of the four estimation methods (three curve-fitting methods and one grouping method). The mean standard dose estimated based on age differed slightly among the curve-fitting methods and was higher at 0.25 years and lower at 15 years using logarithmic or power fitting than using bilinear fitting. Differences between the curve-fitting methods were less apparent for the weight-based estimation.

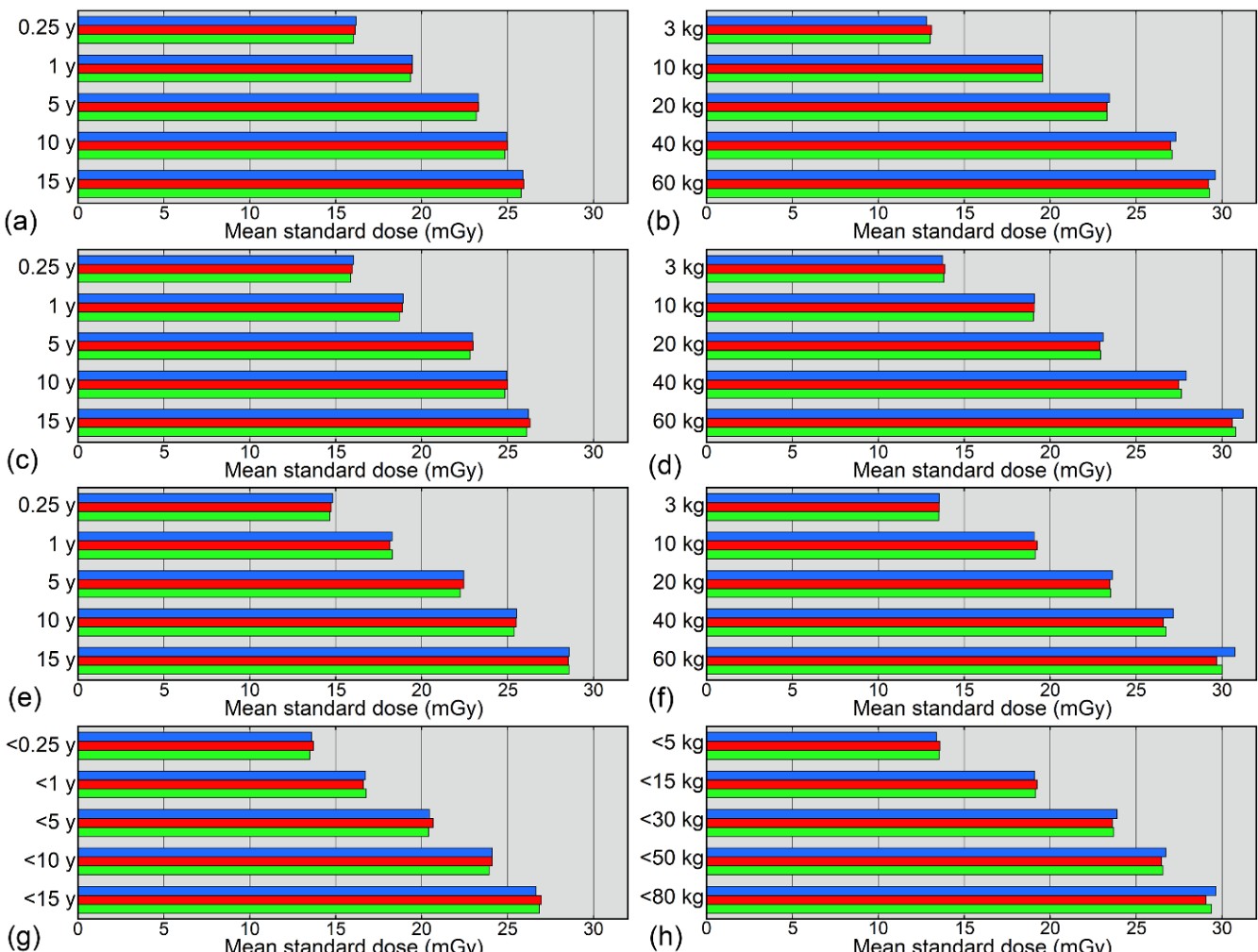

**Figure 4.** Mean standard doses estimated based on age (**a,c,e,g**) or weight (**b,d,f,h**). Standard doses were estimated using logarithmic fitting (**a,b**), power fitting (**c,d**), bilinear fitting (**e,f**), and grouping (**g,h**). The blue, red, and green bars represent values for small, medium, and large datasets, respectively.

The CVs of the standard dose for each age and weight class are presented in Figure 5. Irrespective of the estimation method, the CVs decreased with increasing sample size and tended to be larger for the grouping methods than for the curve-fitting methods. Among the curve-fitting methods, the CVs for the small datasets exceeded 5% at 0.25 years and 3 kg when using logarithmic fitting and at 15 years and 60 kg when using bilinear fitting (Figure 5). Such a large CV was not obtained using power fitting.

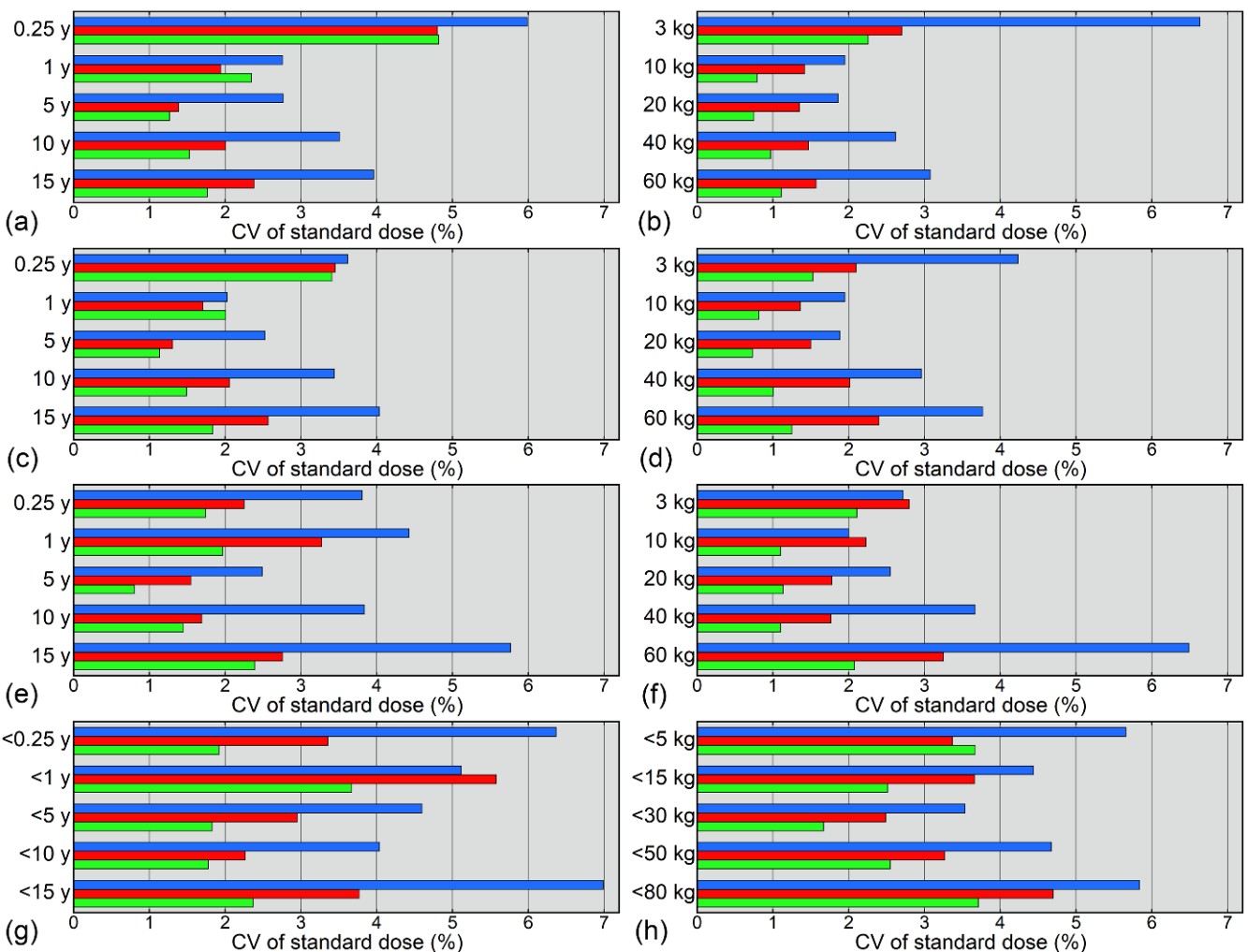

**Figure 5.** CVs of standard doses estimated based on age (**a**,**c**,**e**,**g**) or weight (**b**,**d**,**f**,**h**). Standard doses were estimated using logarithmic fitting (**a**,**b**), power fitting (**c**,**d**), bilinear fitting (**e**,**f**), and grouping (**g**,**h**). The blue, red, and green bars represent values for small, medium, and large datasets, respectively.

## 4. Discussion

In this study, we investigated the methods used to estimate the standard radiation dose for pediatric brain CT at each facility, focusing on the effect of sample size. First, we attempted to optimize the bilinear fitting of the relationships of radiation dose with age and weight. In bilinear fitting, the data for young and old (or light and heavy) children were analyzed separately to obtain two distinct linear regression equations. Since head size increases rapidly soon after birth, preceding the growth of the body [16], the slope of the regression line was lower for the old and heavy ranges than for the young and light ranges, and the slope tended to decrease with increasing cutoff age or weight. Estimation of CTDIvol using monolinear fitting of the dose–age and dose–weight relationships led to severe overestimation at 0–<0.25 years and 0–<5 kg, respectively. Bilinear fitting reduced the errors in the estimation, demonstrating its superiority over monolinear fitting. In the bilinear fitting of the dose–age relationship, the mean error was closest to 0 when the young and old ranges were divided with a cutoff of 1 year. However, the proportion of data at <1 year might be low in general, and lack of data in the young range may prevent valid regression analysis—especially when the entire sample size is small. Considering the small differences in error between the 1- and 1.5-year cutoffs, 1.5 years was selected as the cutoff in the subsequent analysis. For bilinear fitting of the dose–weight relationship, a cutoff of

15 kg was selected, which resulted in successful estimation and equalized the numbers of data in the light and heavy ranges.

For radiation dose management, each facility investigates radiation dose indices recorded in clinical practice and determines the standard dose at the respective facility. The standard dose in pediatric imaging is generally determined as the median value for each age or weight group. However, the number of pediatric CT examinations is small in many facilities, and division into groups further decreases the number in each group, raising concerns about the reliability of standard dose determination. Higher radiation doses for pediatric CT have been reported at non-pediatric facilities than at pediatric facilities [11–13]. It can be inferred that non-pediatric facilities where CT examinations are performed infrequently in children deliver relatively high doses due to lack of experience and knowledge, and that such high dose may not be well-recognized through the current dose surveys. Determination of the standard dose and dose optimization using DRLs are considered important for facilities performing a small number of pediatric CT examinations. Additionally, soon after altering the imaging parameters for optimization, the resultant dose reduction, as well as the acceptability of the image quality, should be assessed, requiring determination of the standard dose using a small sample.

In this study, we evaluated the effect of the sample size on standard dose determination. Random sampling of 25, 50, and 100 CT examinations from 980 examinations was repeated, and the standard dose was then estimated for each dataset. It has been recommended that at least 10 data per age or weight group should be used in determining the median value when establishing DRLs [9], while there are reports where groups with 5 data were included in the analysis [2,20]. In this study, many age or weight groups had fewer than five data when the entire CT dataset consisted of 25 examinations. Increasing the sample size mitigated the lack of data per group; however, for the 50–<80 kg group, fewer than five data were available in 5 of the 20 large datasets consisting of 100 CT examinations. The cutoffs for age and weight grouping differ among DRLs [21,22]. In this study, weight grouping was performed according to the European guidelines [9], although people are smaller in our country than in European countries, which accounts for the lack of data in the large weight group. Grouping may be adjusted to standard body habitus in the country; however, the use of different groupings will hamper international comparisons.

The curve-fitting method has been proposed to deal with the sample size problem in radiation dose management for pediatric body CT [14,15]. In this method, weight is regarded as a continuous variable, instead of grouping, and a curve is fitted to the plots of the radiation dose index against weight. Curve-fitting has also been applied to pediatric brain CT regarding age or weight as a variable [16]. In the present study, standard doses for pediatric brain CT were determined from datasets consisting of 25, 50, and 100 data. The estimation methods used were the grouping and curve-fitting methods, and logarithmic, power, and bilinear functions were applied for fitting. These three functions were selected following a previous study where fitting was performed using all 980 data [16]. The mean standard doses across 20 datasets of a given sample size showed no consistent changes with increasing sample size, regardless of the estimation method, indicating that sample size does not cause systematic errors in standard dose estimates. Although age-based estimation resulted in small differences in the mean standard dose between the curve-fitting methods, such differences were obscured for weight-based estimation. The mean standard doses obtained using age-based grouping were lower than the DRL values reported previously [2,3,9,10,23–25] and similar to [10,25] or lower than [2,23,24] the reported median values. Irrespective of the estimation method, the CVs of the standard dose estimates were relatively large when a small number of data were used, and decreased with increasing sample size. A larger CV implies that the estimated standard dose has a larger random error and is, thus, less reliable. The grouping method tended to yield larger CVs than the curve-fitting methods. To simulate standard dose estimations at facilities performing a small number of pediatric brain CT examinations, a median value was determined even when only one datum was available for a given age or weight

group, which appeared to be related to the large CVs. Smaller CVs for the curve-fitting methods support their effectiveness for determining standard doses. Standard doses can be calculated at any age or weight using the fitting curve, which is expected to aid in comparisons between different DRLs and is considered to be another advantage of the curve-fitting method over the grouping method.

Comparison of different types of fitting function showed that logarithmic and bilinear fittings resulted in large CVs for the young/light groups and old/heavy groups, respectively. The CVs were less conspicuous when using a power function. In a previous study that analyzed a large number of pediatric brain CT examinations [16], agreement between the actual and estimated doses was indicated to be best for bilinear fitting. When the sample size is small, power fitting may be better to suppress random variation and obtain consistent estimates of the standard dose.

In this study, data from a single facility were analyzed, and similar investigations at other facilities should be performed in the future. The relationship between radiation dose and age or weight and, consequently, the optimal fitting function are expected to depend on the imaging method. We used AEC software and modulated the tube current according to X-ray attenuation predicted from lateral localizer images. The AEC software used adjusts the tube current, which is proportional to the radiation exposure, so as to keep the image noise constant [17]. When using AEC, the radiation exposure depends on the type of AEC software, the parameters input into the software, and the direction of the localizer images [26–30]. There are facilities where AEC is not used for pediatric brain CT [10,23]. Without AEC, operators may set the tube current according to the facility's protocol or their personal experiences. Whereas we fixed the tube voltage at 120 kV, lower voltages may be applied in small children [23,31–33]; this may influence dose–age and dose–weight relationships. The optimal fitting function may also depend on the distribution of data across age or weight. It would be recommendable to apply different fitting functions to plots of radiation dose indices against age or weight and visually select fitting functions suitable for the plots.

### 5. Conclusions

In this study, we investigated the estimation of standard radiation doses using a small number of pediatric brain CT examinations. When the sample size was smaller, random variations in the estimated standard dose were larger. Curve-fitting methods allowed better estimation of the standard dose than the grouping method. Power fitting appeared to be more effective than logarithmic and bilinear fitting for suppressing random variation. Determination of standard doses by the curve-fitting method is recommended at facilities where pediatric brain CT is infrequently performed, so as to promote radiation dose optimization nationwide.

**Supplementary Materials:** The following supporting information can be downloaded at: https://www.mdpi.com/article/10.3390/tomography8050207/s1, Figure S1: Examples of plots of CTDIvol against weight for small, medium, and large data sets; Table S1: Coefficients of determination between CTDIvol and age; Table S2: Coefficients of determination between CTDIvol and weight.

**Author Contributions:** Conceptualization, Y.I.; methodology, Y.I. and H.I.; formal analysis, Y.I., H.I., N.S., R.S. and K.M.; investigation, Y.I., H.I., N.S., R.S. and K.M.; resources, Y.I.; data curation, H.I. and R.S.; writing—original draft preparation, Y.I.; writing—review and editing, Y.I., H.I., N.S., R.S. and K.M.; visualization, Y.I.; supervision, Y.I.; project administration, Y.I. All authors have read and agreed to the published version of the manuscript.

**Funding:** This research received no external funding.

**Institutional Review Board Statement:** This study was conducted in accordance with the Declaration of Helsinki and approved by Kitasato University's Medical Ethics Organization (B20-114).

**Informed Consent Statement:** The need for informed consent was waived based on the retrospective study design.

**Data Availability Statement:** The data are available upon reasonable request from the corresponding author.

**Acknowledgments:** We thank Momoko Tatebayashi for her support in data management, Yui Inoue for her support in figure preparation, and Hiroki Miyatake for his advice on statistics.

**Conflicts of Interest:** The authors declare no conflict of interest.

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
