# Peer review of "Sample Size and Estimation of Standard Radiation Doses for Pediatric Brain CT"

_tomography, doi:10.3390/tomography8050207_

Round 1
Reviewer 1 Report
In general, the work is written in a clear language, logically structered and illustrated. However, there are the following remarks about the work:
1. I would like to see a clear and precise rationale for choice of fiiting functions. In paticular, it is logical to assume that at zero weight CTDIvol should tend to zero.
2. For all functions, except for linear ones, there are no parameter values.
3. There is no parameter estimation for all functions. I would also like to see how many points fall within the confidence limits for each functions.
4. To find a bilinear approximation, it is logical to use Chow test with dummy variables and determine the intervals (possible several) in which the linear fitting will be optimally performed. Instead of using "predefined cutoff"
5. I would like to see explanations for Table 7: 1) what is a distribution of CV (normal?) and 2) not just the averages value and errors, as well as the answer to the question - for a given sample size, are the average CV values significantly different for different functions and classes.
Author Response
Comment 1)
I would like to see a clear and precise rationale for choice of fiiting functions. In paticular, it is logical to assume that at zero weight CTDIvol should tend to zero.
Reply 1)
The choice of fitting functions is not logical but empirical. In the previous study (reference 16 in the revised manuscript, Tomography 2022, 82, 985–998), logarithmic, power, and bilinear functions well represented the relationship between CTDIvol and age or weight in all 980 data. In the present study where fitting was performed using a part of the data, we followed the previous study in the selection of the fitting methods.
We added the following comment to the 4th paragraph of Discussion:
“These three functions were selected following the previous study where fitting was performed using all the 980 data [16].”
Regarding the zero point problem, we agree the reviewer’s theoretical view. Again, discrepancy of the y-intercept from zero is derived from the empiric nature of analysis. We think that such empiric nature is usual and rather inevitable for radiation dose management. Both ICRP (publication 135) and European Commission (Radiation Protection 185) recommended use of exponential curve fitting for body CT without a constant. The expected dose index is not zero at weight of 0 kg.
Comment 2)
For all functions, except for linear ones, there are no parameter values.
Reply 2)
For the assessment of “Age and Weight Division for Bilinear Fitting”, fitting was performed once for each condition (age/weight, cutoff), and one regression line and one correlation coefficient were determined for each condition. The relationship between age or weight range and the slope of the regression line appear to have significant information. Thus, we presented the regression equation and correlation coefficient.
In contrary, for the assessment of “Sample Size and Standard Dose Estimation”, we performed fitting for 20 datasets per condition (age/weight, type of fitting function, sample size). In the previous study, the equations for the three types of fitting function and for age and weight were presented (Table 1 in reference 16 in the revised manuscript). However, presentation of the results from 20 datasets for each sample size (25, 50, and 100 data) does not seem to be beneficial. The final purpose of fitting was to determine the standard doses at designated ages and weights. Thus, we presented the means and coefficient of variations of the estimated standard dose to demonstrate the quality of estimation.
Comment 3)
There is no parameter estimation for all functions. I would also like to see how many points fall within the confidence limits for each functions.
Reply 3)
Regarding the quality of fitting, we added the following comments and supplementary tables:
“The sample size did not affect the median values of the coefficients of determination between CTDIvol and age (Table S1) or weight (Table S2) whereas the coefficients of determination were exceptionally low in one small dataset.”
Because reduction of Figures and Tables were requested by another reviewer, we presented the new tables as supplementary materials.
Comment 4)
To find a bilinear approximation, it is logical to use Chow test with dummy variables and determine the intervals (possible several) in which the linear fitting will be optimally performed. Instead of using "predefined cutoff"
Reply 4)
Again, we agree that our analysis is empirical and practical rather than statistically logical. We selected simple values (1, 1.5, and 2 years for age, and 10, 15, and 20 kg for weight) as candidates of the single cutoff, considering convenience for wide use in many facilities, and evaluated systematic overestimation and underestimation in each age or weight group. We attempted to find a practical method to estimate the standard dose with acceptable accuracy for each age or weight group.
Comment 5)
I would like to see explanations for Table 7: 1) what is a distribution of CV (normal?) and 2) not just the averages value and errors, as well as the answer to the question - for a given sample size, are the average CV values significantly different for different functions and classes.
Reply 5)
As for the age-based estimation by the curve-fitting method, for example, we averaged CV values at 0.25, 1, 5, 10, and 15 years. We supposed that Table 7 in the original manuscript would aid recognition of CV differences according among the estimation methods. However, the implication of this calculation may be vague, and we realized that it may be confusing. Thus, we deleted the table. We appreciate the reviewer’s comment.
Reviewer 2 Report
Radiation doses in pediatric population represents a long lasting problem not only in radiology but also in other subspecialisations of medicine. The article is interesting, but the general conceptualisation must be improved.
I can barely describe in details what exactly to be changed.
I would suggest:
Reduce the number of figures
Reduce the number of tables
Improve abstract
Improve resuts section
Make the article easier to read and understand.
Simplify main ideas
The section which I consider adequate are:
Introduction and discussion and references.
Author Response
We agree that the numbers of figures and tables are large.
We moved Figure 4 in the original manuscript to the supplementary file.
Another reviewer suggested that Table 7 is difficult to understand, and therefore we deleted it.
Two tables were created according to the reviewer’s comment, but they were presented as supplementary materials.
As for the Results section, additionally, we deleted the first two sentences in the first paragraph of “3.2. Sample Size and Standard Dose Estimation”. The two sentences presented the numbers in the age and weight groups, which were also shown in Tables 3 and 4. We added the following comment associated with the numbers, to the last of “3.1. Age and Weight Division for Bilinear Fitting”.
“Additionally, Table 4 shows that the number of examinations for the 50 – < 80 kg weight group was relatively small.”
We believe that these revisions would improve the readability of the Results section.
The main purpose was to improve optimization of radiation dose in pediatric brain CT. We revised the last part of Abstract and Introduction to clarify this goal.
(Abstract, before revision)
“It is recommended to determine the standard dose for pediatric brain CT by the curve-fitting method at facilities performing the imaging procedure infrequently. ”
(Abstract, after revision)
“Determination of the standard dose for pediatric brain CT by the curve-fitting method is recommended at facilities performing the imaging procedure infrequently to improve radiation dose optimization. ”
(Introduction, before revision)
“The principal aim of this study was to aid standard dose determination for pediatric brain CT at facilities where this imaging procedure is infrequently performed. ”
(Introduction, after revision)
“The principal aim of this study was to aid standard dose determination and radiation dose optimization for pediatric brain CT at facilities where this imaging procedure is infrequently performed. ”
Moreover, we added the term "diagnostic reference levels" to the 1st sentence of Abstract to indicate our study is related to the diagnostic reference levels.
(before revision)
“Estimation of the standard radiation dose at each imaging facility is required for radiation dose management.”
(after revision)
“Estimation of the standard radiation dose at each imaging facility is required for radiation dose management including establishment and utilization of the diagnostic reference levels.”
Reviewer 3 Report
Start introduction with Ct doses and their effects then go to DRL and recommendations from ICRP as well as other organizations.
Cite some papers from current journal
compare well your results with recent published paper and DRL in other countries or even local LDRL.
Author Response
Comment 1)
Start introduction with Ct doses and their effects then go to DRL and recommendations from ICRP as well as other organizations.
Reply 1)
According to the reviewer’s comment, we moved the following description to the 1st paragraph (reference numbers has been changed):
“Computed tomography (CT) is the major source of radiation exposure. Children are sensitive to radiation, and their expected long lifetimes allow cancer development after a long latency period [1]. Accordingly, optimizing the radiation dose for pediatric CT is a priority.”
“In children, CT is most frequently used for brain imaging [2,3], and an increased incidence of brain tumors has been reported in children who underwent brain CT [4–6]. Therefore, optimization of the radiation dose in pediatric brain CT is of particular im-portance.”
This revision should clarify that our root motivation was to improve optimization of pediatric brain CT. We appreciate the reviewer’s suggestion.
Comment 2)
Cite some papers from current journal compare well your results with recent published paper and DRL in other countries or even local LDRL.
Reply 2)
We compared our standard doses with the previous papers describing DRLs, and added the following sentence to the 5th paragraph of Discussion:
“The mean standard doses obtained using age-based grouping were lower than the DRL values reported previously [2,3,9,10,23–25] and similar to [10,25] or lower than [2,23,24] the reported median CTDIvol values.”
References 24 and 25 were added in this revision.
Round 2
Reviewer 2 Report
The article was considerably changed in respect of rewivers comments. The readibility and scientific soundness now was improved and meet high quality criteria now. I feel that the article might be accepted in this form.